# Fast Attention CNN for Fine-Grained Crack Segmentation

**DOI:** 10.3390/s23042244

**Published:** 2023-02-16

**Authors:** Hyunnam Lee, Juhan Yoo

**Affiliations:** 1Incheon International Airport Corporation, Incheon 22382, Republic of Korea; 2Department of Computer, Semyung University, Jecheon 02468, Republic of Korea

**Keywords:** crack detection, convolutional neural network, image segmentation, salient object detection

## Abstract

Deep learning-based computer vision algorithms, especially image segmentation, have been successfully applied to pixel-level crack detection. The prediction accuracy relies heavily on detecting the performance of fine-grained cracks and removing crack-like noise. We propose a fast encoder-decoder network with scaling attention. We focus on a low-level feature map by minimizing encoder-decoder pairs and adopting an Atrous Spatial Pyramid Pooling (ASPP) layer to improve the detection accuracy of tiny cracks. Another challenge is the reduction in crack-like noise. This introduces a novel scaling attention, $AG^{+}$, to suppress irrelevant regions. However, removing crack-like noise, such as grooving, is difficult by using only improved segmentation networks. In this study, a crack dataset is generated. It contains 11,226 sets of images and masks, which are effective for detecting detailed tiny cracks and removing non-semantic objects. Our model is evaluated on the generated dataset and compared with *state-of-the-art* segmentation networks. We use the mean Dice coefficient (mDice) and mean Intersection over union (mIoU) to compare the performance and FLOPs for computational complexity. The experimental results show that our model improves the detection accuracy of fine-grained cracks and reduces the computational cost dramatically. The mDice score of the proposed model is close to the best score, with only a 1.2% difference but two times fewer FLOPs.

## 1. Introduction

With the excellent achievements of convolutional neural network (CNN) based computer vision algorithms, object detection and image segmentation have been widely used in various research areas, such as surveillance systems, autonomous driving, and medical image analysis. An object detection algorithm provides category and location information of objects within an image. Image segmentation is a pixel-wise classification process. Both these algorithms have been successfully applied for automated pavement defect inspection. Image segmentation approaches provide more information than object detection for the detected pavement defects.

When object detection algorithms are applied to inspect defects, both the location and type of defects are achieved simultaneously. Multi-stage approaches are used in case of image segmentation. It consists of three steps: (1) cracks are detected using CNN-based image segmentation models, (2) defects are estimated by grouping the detected cracks, and (3) the type and severity level of defects are predicted. Thus, image segmentation-based crack detection is a fundamental research area in pavement defect inspection. In this study, we present an image segmentation-based crack detection algorithm to handle the defect inspection problem.

Although deep learning-based approaches have dramatically improved crack detection accuracy, crack detection is still challenging. In contrast to semantic scene segmentation, crack detection accuracy highly depends on detecting fine-grained cracks and separating cracks from crack-like noise, such as grooving, skid marks, and road lanes.

Encoder-decoder architectures, such as U-Net [1] and SegNet [2], are commonly used in crack segmentation. To achieve more accurate results in image segmentation, the deep learning architecture is improved by varying depth and width (Double U-Net [3]), introducing new types of modules (ASPP [4], attention module [5]), and adopting new architectures (vision-transformer [6]). However, improved models do not always outperform on crack segmentation because they are not effective in detecting tiny cracks. Although improved architectures provide rich contextual features, they result in the loss of spatial information from low-level feature maps. Deep learning models with deeper and larger architectures [7] and multi-scale aggregated feature maps [8] provide better representation power for fine-grained crack detection, but they are computationally expensive. A higher computational cost is required to maintain the spatial information from low-level feature maps. Considering the large amount of pavement inspection data generated in practice, reducing computational cost is essential. We propose an encoder-decoder architecture model to reduce the computational cost significantly with performance improvement.

Another limitation is removing crack-like noise. Although many attention-based models exhibit outstanding performance in highlighting salient regions and suppressing irrelevant areas, crack segmentation approaches still suffer from hard negatives, such as grooving and skid marks. Development of segmentation models, as well as a training dataset, is a vital factor in improving segmentation accuracy, particularly in removing crack-like noise. In addition, special purpose image segmentation datasets (e.g., crack dataset and polyp detection dataset) do not have sufficient pixel-level labeled datasets, unlike semantic segmentation datasets, such as COCO [9]. Therefore, we generate a crack dataset that includes various cracks in diverse pavement environments and sufficient hard negatives.

In this study, we present a simple and effective crack segmentation model. The proposed model has an encoder-decoder architecture with scaling attention. It focuses on low-level features to simultaneously improve the detection accuracy of tiny cracks and computational effectiveness. We introduce a new scaling attention method, Attention Gate${}^{+}$ (AG${}^{+}$), and a modified convolution block in decoders and encoders to deal with crack-like noise. In addition, we verify that our own dataset is effective in eliminating hard negatives. Our contributions are summarized as follows:A new scaling attention module, Attention Gate${}^{+}$ (AG${}^{+}$), in decoders is proposed. It generates a coefficient vector by combining encoder’s skip connection and decoder’s feature map to highlight salient regions.We propose a simple encoder-decoder architecture to achieve spatial information from low-level feature maps and computational effectiveness simultaneously. Modified convolutional blocks in encoders and decoders are used to reduce the number of encoder-decoder pairs. This results in accuracy improvement and computational efficiency. The proposed model reduces the computational cost by 51% (FLOPs) when compared with the outperformed segmentation model, but the accuracy difference is only 1.2% (mDice).We generate our own crack dataset that contains balanced cracks and sufficient hard negative data in various environments. Our crack dataset is verified to improve performance in detecting fine-grained cracks and removing crack-like noise through experiments.

The rest of this paper is organized as follows. We first provide a brief review of the related work in Section 2. In Section 3, the overall architecture of the proposed model and its components are defined. Section 4 describes experiments and results. We then present our conclusion in Section 5.

## 2. Related Work

Automated pavement defect detection approaches can be categorized into two groups: object detection [10,11,12,13] and pixel-wise segmentation [6,14,15,16,17]. Early studies adopted object detection networks to localize cracks owing to significant progress in object detection networks. However, the bounding box does not provide sufficient information about cracks, such as their degree and shape. Pixel-wise crack detection is required to achieve precise crack shape and length.

Crack detection is closer to Salient Object Detection (SOD) than semantic segmentation [2], because it focuses on separating cracks from the background. Since U-Net [1] has achieved significant progress in image segmentation, the encoder-decoder architecture is the most common network in image segmentation-based crack detection. U-Net aggregates multi-level feature maps using skip connections and incremental upsampling. After the success of U-Net in biomedical segmentation, the U-Net-like SOD network has been introduced in polyp segmentation. These networks are now used in various research fields, such as crack detection. Recent SOD algorithms with encoder-decoder structures adopt the attention method to enhance accuracy. PraNet [18] and CaraNet [19] use reverse attention and a partial decoder. UACANet [20] introduced an uncertainty area and parallel axial attention. Attention U-Net [21] suggested a scaling attention module, Attention Gate, instead of the attention module to alleviate the computational cost. Attention Gate estimates salient regions using a coefficient grid vector.

Another category of U-Net-like models modifies the encoder-decoder structure. Double U-Net [3] combines two modified encoder-decoder networks in sequence, and it improves U-Net structure using ASPP [4] and Squeeze and Excitation (SE) block [22]. Chen and Lin [17] presented a crack detection network using multiple ASPP concatenations called HACNet. Liu et al. [23] proposed DeepCrack which generates pyramid feature maps and concatenates every level feature maps. Another DeepCrack [8] also combines all scale feature maps, but each scale feature map is generated by fusing the skip connection of encoder and the feature map of decoder at each scale. SSDNet [24] proposes the DenSep module to generate multi-scale feature maps. DenSep modules use a point-wise and a depth-wise convolution to reduce the computational cost. Lau et al. [25] presented the U-Net based crack detection network that replaces encoders with ResNet-34.

After the vision transformer was introduced in [26], the transformer plays a key role in increasing segmentation performance. Vision transformers outperform CNN in terms of spatial information and global feature maintenance [27]. In SSFormer [28], the encoder is replaced with a vision transformer. In [28], a performance comparison was conducted with various vision transformers, such as CvT [29], PvT [30], and MiT [31]. MiT shows the best results because it is the best method for preserving spatial information. CrackFormer [6] presents crack segmentation model based on a vision transformer. In [6], a modified scaling attention module was used to obtain a crisp crack boundary map.

In case of crack detection, CNN-based models present better results than vision transformer-based models. Crack detection requires pixel-level segmentation information, whereas vision transformers use image patches. A Hybrid model which integrates strengths of CNN and vision transformer is proposed to address limitations of vision transformers. FCBformer [32] fuses PvT transformer with a fully convolutional network. FCBformer estimates feature maps from input images instead of low resolution to overcome the limitation of the hybrid model.

Segmentation accuracy depends on input size, particularly for small objects. The larger input size, the higher the computational complexity. The hybrid model must be more expensive with a larger input. Therefore, the proposed model adopts a CNN-based architecture with a larger input. To improve the segmentation accuracy, we applied diverse modules, such as the new scaling attention module, ASPP, SE block.

## 3. Architecture

### 3.1. Overall Architecture

The proposed crack segmentation model adopts encoder-decoder structures. As shown in Figure 1, the fundamental components of encoder and decoder are based on Double U-Net [3]. The overall architecture is an encoder-ASPP-decoder. Our research focuses on recall rather than precision. Accuracy drop results from False Negative rather than False Positive because cracks have line shape and small sparse property. We use only three pairs of encoder-decoder to enhance the fine-level feature map. The ASPP layer is effective in capturing multi-scale context information.

The proposed model utilizes a scaling attention module that emphasizes a salient area with low computational costs to deal with crack-like noise. We introduce a novel scaling attention module called Attention Gate${}^{+}$(AG${}^{+}$). Another method for noise reduction is convolution block3 (CB3). We modify convolution blocks in both encoder and decoder. Figure 2 shows the structures of the AG${}^{+}$ and CB3.

### 3.2. Attention Gate${}^{+}$(AG${}^{+}$)

Attention Gate${}^{+}$(AG${}^{+}$) is inspired by the Attention Gate in Attention U-Net [21]. In decoder, AG${}^{+}$ becomes a mask to highlight salient features. AG${}^{+}$ generates a coefficient vector by using a skip connection from each encoder layer and a feature map from each decoder layer. In [21], Attention Gate was introduced to detect the area shape of the pancreas, but we aim to segment line-shaped cracks. AG${}^{+}$ is proposed using an additional convolution to improve segmentation performance for line features. Although it requires extra computational overhead, it results experimentally in better accuracy in line shape segmentation. The left-hand side of Figure 2 shows the novel scaling attention AG${}^{+}$. The coefficient vector multiplies element-wise with the feature map in each decoder layer as shown in Figure 1 and Figure 2.

### 3.3. Encoder and Decoder

We adopt a fundamental encoder and decoder structure from Double U-net [3]. Encoder and decoder are based on a convolution block consisting of 3 × 3 convolution, batch normalization, ReLU (Rectified Linear Unit) activation function. Encoder employs max-pooling, and decoder employs upsample to resize spatial dimensions of feature maps. An SE block is placed after convolution block for channel attention effect. We improve the convolution block, called CB3, with an additional 3 × 3 convolution, batch norm, and ReLU shown in the right-hand side of Figure 2. CB3 is helpful to overcome the lack of channels and convolution layers.

Output is generated through a 1 × 1 convolution layer with a Sigmoid activation function.

## 4. Experimental Results

### 4.1. Dataset

Open datasets for cracks are not sufficient for training deep learning models. Most open datasets have 100∼500 images. Although the SDNET2018 [33] dataset contains 56,000 images, it includes cracks in both pavement and other areas, such as buildings and bridge decks. Moreover, the crack areas in the images are at most 6%, usually less than 3% [17]. We generate our own crack dataset to obtain sufficient and balanced crack data.

In terms of detailed tiny cracks and hard negative data, the public crack dataset is not sufficient. In practice, fine-grained cracks are the most common defects, and hard negatives are fatal factors that decrease accuracy. Our crack dataset focuses on tiny long cracks and hard negative data to enhance performance.

The crack dataset is generated using digital images from the pavement. We use two laser line scanners mounted on a vehicle to capture crack images. The depth of cracks and hard negative data, such as skid marks, grooving and road lanes are captured by top view. These 3D data are transformed into 2D crack images, and crack areas are labeled manually. It contained 11,226 sets of images and masks.

Each image contains 4 m × 4 m road area. There are two types of image size, 512 × 512 and 1024 × 1024, to handle the crack size change according to the image resolution. One pixel in the images is about 4 mm in case of 1024 × 1024 image resolution and 8 mm in 512 × 512. Therefore, our dataset can illustrate fine-grained cracks which are less than 1 cm. Most of the crack widths are less than 10 pixels. The crack area rate in our dataset is approximately 1%. Our dataset provides sufficient crack data to train the model because it has 100× more images than open crack datasets. To reduce False Positive, the crack dataset contains various types of hard negative images. Data rotation is used to deal with all directional cracks and hard negatives. Images in the first column of Figure 4 present typical cases of our crack dataset.

The proposed and comparison models are trained and evaluated on our own created crack dataset.

### 4.2. Evaluation Metrics

mDice and mIoU are used to evaluate our model and compare it with other segmentation models. mDice and mIoU are popular metrics for evaluating segmentation algorithms. Both metrics compute the ratio between the intersection and the union of ground truth (GT) and prediction in pixels. Dice coefficient and IoU are
(1)$$\frac{2\times (P\cap GT)}{\left|P|+|GT\right|}$$
(2)$$\frac{P\cap GT}{P\cup GT}$$
where *P* is prediction and *GT* is ground truth. Because crack data are usually thin and long line shapes, metrics do not show high scores, even though prediction is very similar to *GT* in Figure 3. A small difference in pixels results in a large difference in score because the number of pixels of *GT* and prediction is small. The average crack area rate is approximately 1% of the input images, and the average width of cracks is less than 10 pixels, which is less than 2% of the image width. mDice and mIoU are essential to show how much the accuracy has been improved even though prediction can hardly estimate the same coordinate to *GT* exactly.

Figure 3 presents a comparison between *GT* and Prediction according to the dice coefficient score: (a) is one of the highest scores and (b) is close to mDice of our model and comparison models. In case (a), the intersection area (white pixels) seamlessly illustrates the crack. In case (b), the intersection area is scattered on the crack line with more False Negative. Although the result of (b) does not show a seamless prediction, it provides sufficient information to estimate the crack shape and location.

### 4.3. Implementation Details

We use 512 × 512 as the input size for both training and inference to detect tiny cracks. We use the Adam optimizer [34] and dice-loss. The learning rate is set to $10^{-4}$, the batch size to 8, and epoch to 300. Table 1 summarizes hyper-parameters used during training.

These hyper-parameters affect the local optima and prediction accuracy. We attempt a few options and decide the optimal hyper-parameters set. Step Learning rate, learning rate warmup, and fixed learning are tried. We attempt 2, 4, 8 as the batch size and SGD, Nadam, Adam for the optimizer. The selected hyper-parameter set shows the best prediction accuracy. Dice-loss is the most popular loss function for image segmentation. It is effective to train our model because the training loss curve and validation loss curve decrease to a stable point with the number of epochs.

The proposed model is implemented using Tensorflow 2.6 [35] and a single RTX 3090 GPU for training. Comparison models are trained using their original code and hyper-parameters, except for the input size. We modified the input size and related parameters to 512 × 512.

### 4.4. Result

We compare the proposed method against *state-of-the-art* methods on our own dataset. Four different types of models are selected: U-Net, UACANet, Double U-Net, and FCBFormer. U-Net is the most common crack segmentation method. The others show best performance in their categories: UACANet in attention-based models, FCBFormer in vision transformer-based models, Double U-Net in encoder-decoder structure networks. Test images are 207 sets of images and masks that contain various types of cracks, hard negatives, and pavement conditions.

The results of crack detection are given in Figure 4. As we mentioned in Section 4.2, mDice and mIoU do not show good score because of the property of our dataset. Most of the cracks in our dataset are fine-grained cracks, so a small difference in prediction and GT causes a large difference in mDice and mIoU. However, mDice and mIoU scores are sufficient to present the accuracy of the crack segmentation models. Although mDice is around 0.6, the prediction results provide sufficient information for cracks, such as level of severity, direction, length, and shape. The difference in performance depends on how well thin cracks can be detected. In the first row, all methods can detect thick distinct cracks, but the accuracy difference is from the detailed thin cracks. UACANet exhibits the worst performance in thin crack detection among the compared models because it focuses on high-level semantic features using a partial Axial Attention Decoder. U-Net and UACANet have limitations to capture fine-grained cracks. FCBFormer and our model outperform on detecting the detailed tiny cracks.

Because the training dataset includes various hard negative data, all methods perform well in crack detection in noisy environments, such as grooving, skid marks, and road lanes. In the case of grooving, which is the most common crack-like noise on pavements, our model and FCBFormer present better segmentation performance. Experimental results address that our training dataset is more effective in dealing with hard negatives.

Table 2 and Figure 5 illustrate the relationship between performance and computational effectiveness. The proposed model shows the second highest mDice and mIoU scores. Our method has a score close to that of the best model, FCBFormer. The difference is only 1.2% for mDice and 2% for mIoU. However, our model decreases FLOPs by 51% compared with FCBFormer. mDice scores of FCBFormer and our model are 0.585 and 0.578, respectively. FLOPs of FCBFormer and our model are 301 G and 147 G. Our model outperforms the model with similar FLOPs, UACANet. A 16% gap in FLOPs results in a 0.2392 mDice score gap. mDice scores and FLOPs of UACANet are 0.3388 and 126 G.

### 4.5. Ablation Study

We also conduct an ablation study to evaluate the effectiveness of each module in the proposed model. Table 3 summarizes the prediction accuracy according to module change. We select the encoder-ASPP-decoder model without the scaling attention module and additional convolution block as the base model. Experimental results show that AG${}^{+}$ is better than the original Attention Gate (AG) for crack detection. mDice scores with AG module and AG${}^{+}$ module are 0.5536 and 0.5595.

The base model uses 12 convolutional blocks in decoders and encoders. When CB3 is applied, 6 more convolutional blocks are added. Five more convolutional blocks are added for insertion of the AG${}^{+}$ module. Table 3 shows the increment of FLOPs according to module change.

mDice scores increase 2.3% with AG${}^{+}$ and 2.8% with CB3. CB3 is more effective than AG${}^{+}$, but CB3 causes more computational cost. When AG${}^{+}$ and CB3 are applied, mDice and mIoU increase by 5.7% and 7.1%, respectively. When both AG${}^{+}$ and CB3 are used simultaneously, the increment of mDice score is higher than the sum of the increment of AG${}^{+}$ and CB3. These results indicate that AG${}^{+}$ and CB3 create synergistic effects.

## 5. Conclusions

We present a simple and effective crack segmentation network with a novel scaling attention module, AG${}^{+}$. For prediction accuracy, our model focuses on low-level feature maps by minimizing encoder-decoder pairs and inserting an ASPP layer. Scaling attention, modified convolution block, and SE block are applied to deal with crack-like noise while preserving computational efficiency. The effectiveness of AG${}^{+}$ and CB3 is identified through an ablation study. Each module can be easily applied in an image segmentation model to boost prediction accuracy. We also generate a crack dataset comprising 11,226 sets of images and masks. This provides balanced crack data and sufficient hard negative data.

Through experiments, the proposed model shows a high representation power with a low computational cost. When compared with *state-of-the-art* segmentation models, mDice difference is only 1.2%, but FLOPs are reduced by 51%. We also demonstrate that our own crack dataset is effective for removing crack-like noise.

The proposed model focuses on detecting the fine-grained cracks, so the test dataset consists of detailed thin cracks. Although experimental results illustrate that prediction provides sufficient information for cracks, mDice scores of the proposed and the comparison models are not good. In the future, we aim to develop the proposed model to enhance prediction accuracy in all type of cracks, such as fine-grained and thick cracks. In addition, we would like to improve AG${}^{+}$ and verify its effectiveness through applying it to image segmentation models.

## Figures and Tables

**Figure 1 sensors-23-02244-f001:**
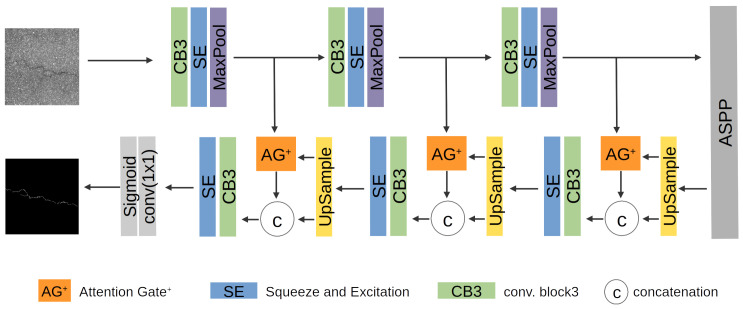
Architecture of the proposed model.

**Figure 2 sensors-23-02244-f002:**
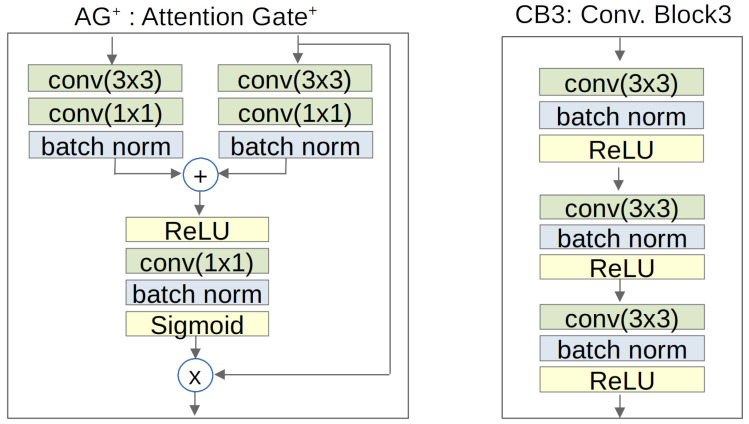
Attention Gate${}^{+}$ (AG${}^{+}$) and convolution block3 (CB3).

**Figure 3 sensors-23-02244-f003:**
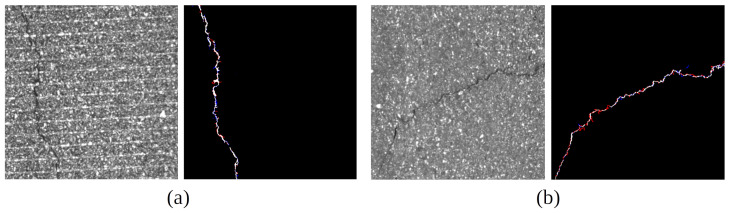
Comparison with *GT* and our prediction at different accuracies. White pixels present intersection between *GT* and our prediction. Red pixels are *GT*, and blue pixels are crack segmentation results, respectively. (**a**) Dice is 0.83, (**b**) dice is 0.58.

**Figure 4 sensors-23-02244-f004:**
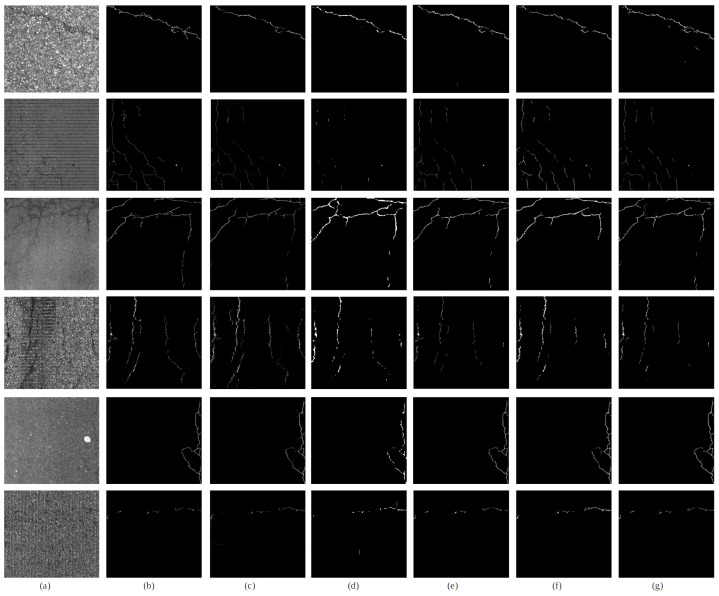
Prediction comparison with other segmentation models. (**a**) Pavement images, (**b**) ground truth, (**c**) U-Net, (**d**) UACANet, (**e**) Double U-Net, (**f**) FCBFormer, (**g**) Ours. Most models can detect distinct cracks as seen in the first row. The performance difference is from the fine-grained cracks. U-Net and UACANet tend to miss tiny cracks, but FCBFormer and our model outperform on detecting fine-grained cracks.

**Figure 5 sensors-23-02244-f005:**
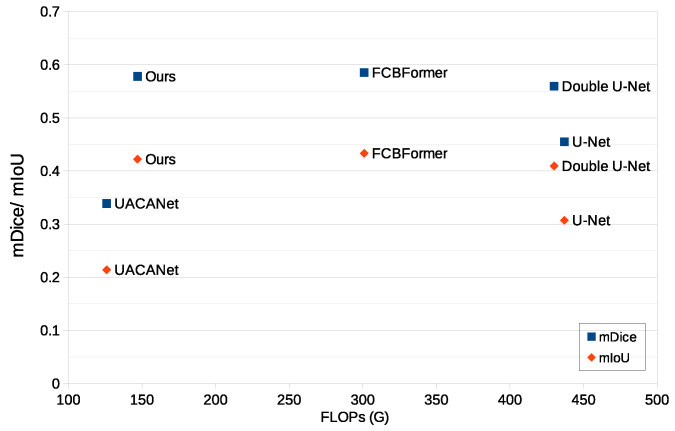
Comparison of performance and computational complexity.

**Table 1 sensors-23-02244-t001:** Hyper-parameters.

Parameters	Batch Size	Learning Rate	Epoch	Optimizer	Loss
values	8	$10^{-4}$	300	Adam	dice-loss

**Table 2 sensors-23-02244-t002:** Performance and computational complexity comparison to *the state-of-the-art* and ours.

Model	mDice	mIoU	FLOPs (G)
U-Net	0.4550	0.3071	437
UACA-Net	0.3388	0.2139	126
FCBFormer	0.5850	0.4332	301
Double U-Net	0.5594	0.4093	430
Ours	0.5780	0.4222	147

**Table 3 sensors-23-02244-t003:** Evaluate the effectiveness of each module in the proposed model.

Model	mDice	mIoU	FLOPs (G)
base model	0.5467	0.3943	94
+AG	0.5536 (+1.2%)	0.3996 (+1.4%)	75 (−2%)
+AG${}^{+}$	0.5595 (+2.3%)	0.4058 (+2.9%)	118 (+26%)
+CB3	0.5618 (+2.8%)	0.4062 (+3.0%)	124 (+32%)
+AG${}^{+}$ & CB3	0.5780(+5.7%)	0.4221 (+7.1%)	147 (+56%)

## Data Availability

Not applicable.

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
