# Peer review of "Fast Attention CNN for Fine-Grained Crack Segmentation"

_sensors, 2023, doi:10.3390/s23042244_

Round 1
Reviewer 1 Report
The authors proposed a model for detecting fine-grained cracks. Comparing with the state-of-the-art methods, their method has almost the top accuracy as well as highest efficiency. This indicates that the newly developed model is more advanced than the conventional methods. The authors performed a good work. I suggest to publish it in the present form.
Author Response
- The authors proposed a model for detecting fine-grained cracks. Comparing with the state-of-the-art methods, their method has almost the top accuracy as well as the highest efficiency. This indicates that the newly developed model is more advanced than the conventional methods. The authors performed good work. I suggest publishing it in the present form.
Author response
Thank you for your insightful review.

Reviewer 2 Report
In this paper, authors introduce a deep learning-based crack segmentation method and a crack dataset. Experimental results show that the proposed method has comparable performance to state-of-the-art methods on the crack dataset, but with lower computational complexity. I have the following concerns:
1. Some work related to crack segmentation is missing and should be cited or compared in the manuscript:
(1) Choi, W., & Cha, Y. J. (2019). SDDNet: Real-time crack segmentation. IEEE Transactions on Industrial Electronics, 67(9), 8016-8025.
(2) Liu, Y., Yao, J., Lu, X., Xie, R., & Li, L. (2019). DeepCrack: A deep hierarchical feature learning architecture for crack segmentation. Neurocomputing, 338, 139-153.
(3) Lau, S. L., Chong, E. K., Yang, X., & Wang, X. (2020). Automated pavement crack segmentation using u-net-based convolutional neural network. IEEE Access, 8, 114892-114899.
2. The authors only present results on the their crack dataset but not on public datasets. To verify the generalization capability of the proposed method, I suggest conducting experiments on existing public datasets, such as the CFD dataset [1] or the Crack500 [2] dataset.
[1] Shi, Y., Cui, L., Qi, Z., Meng, F., & Chen, Z. (2016). Automatic road crack detection using random structured forests. IEEE Transactions on Intelligent Transportation Systems, 17(12), 3434-3445.
[2] Yang, F., Zhang, L., Yu, S., Prokhorov, D., Mei, X., & Ling, H. (2019). Feature pyramid and hierarchical boosting network for pavement crack detection. IEEE Transactions on Intelligent Transportation Systems, 21(4), 1525-1535.
3. Ablation study should be improved:
(1) The number of convolutional layers in the encoder-decoder architecture should be discussed.
(2) In Figure 2, the attention gate+ does not use a nonlinear activation unit between the conv(3x3) and conv(1x1) modules. I suggest conducting experiments to analyze the difference with and without nonlinear activation units.
4. Some writing problems should be avoided:
(1) There should be no spaces before “where” in line 209.
(2) There should be no brackets in the reference [33].
Author Response
- Some work related to crack segmentation is missing and should be cited or compared in the manuscript:
(1) Choi, W., & Cha, Y. J. (2019). SDDNet: Real-time crack segmentation. IEEE Transactions on Industrial Electronics, 67(9), 8016-8025.
(2) Liu, Y., Yao, J., Lu, X., Xie, R., & Li, L. (2019). DeepCrack: A deep hierarchical feature learning architecture for crack segmentation. Neurocomputing, 338, 139-153.
(3) Lau, S. L., Chong, E. K., Yang, X., & Wang, X. (2020). Automated pavement crack segmentation using u-net-based convolutional neural network. IEEE Access, 8, 114892-114899.
Author response
We added the related works that you recommended.
- The authors only present results on the their crack dataset but not on public datasets. To verify the generalization capability of the proposed method, I suggest conducting experiments on existing public datasets, such as the CFD dataset [1] or the Crack500 [2] dataset.
[1] Shi, Y., Cui, L., Qi, Z., Meng, F., & Chen, Z. (2016). Automatic road crack detection using random structured forests. IEEE Transactions on Intelligent Transportation Systems, 17(12), 3434-3445.
[2] Yang, F., Zhang, L., Yu, S., Prokhorov, D., Mei, X., & Ling, H. (2019). Feature pyramid and hierarchical boosting network for pavement crack detection. IEEE Transactions on Intelligent Transportation Systems, 21(4), 1525-1535.
Author response
We deeply agree with your comments. Experiments with public crack dataset is necessary process. Mentioned in section 4.1 Dataset, our paper focuses on the fine-grained cracks and public dataset is not sufficient to train our model, so we generated our own crack dataset specialized in the fine-grained cracks.
We would like to emphasize that the proposed network show the excellent performance with the low computational cost in detecting fine-grained cracks. Our own dataset is good test set for fine-grained cracks.
We are going to conduct comparison experiments with public crack datasets to show generalization of the crack detection model in the future work.
- Ablation study should be improved:
- The number of convolutional layers in the encoder-decoder architecture should be discussed.
Author response
We mentioned the number of convolutional layers according to module changes.
(2) In Figure 2, the attention gate+ does not use a nonlinear activation unit between the conv(3x3) and conv(1x1) modules. I suggest conducting experiments to analyze the difference with and without nonlinear activation units.
Author response
We also think the suggest experiment is important. However, the purpose of ablation study in our works is to address the effectiveness of each module in the proposed crack network. We decided that the scope of ablation study is to show prediction accuracy according to addition of modules (AG+, CB3) without any component modification.
Your suggestion is excellent idea for development of "the attention gate+ module". We are going to conduct comparison experiments to find new crack detection model in the future work.
- Some writing problems should be avoided
(1) There should be no spaces before “where” in line 209.
(2) There should be no brackets in the reference [33].
Author response
We modified writing problems that you mentioned.

Reviewer 3 Report
This paper had presented a simple and effective crack segmentation network with a novel scaling attention module.
Pros:
1. The paper is easy to read and well organized, making it easier for readers to follow the authors idea and get a better understanding.
2. Sufficient figures are provided to make the results more convincing.
Cons:
1. Include a separate table mentioning the hyper-parameter settings of the proposed model.
2. Mention the details pertaning to the model parameter uncertainty of the proposed model.
3. Discuss about the dice score stability during the training of the proposed model.
4. Make sure your conclusions appropriately reflect on the strengths and weaknesses of your work, how others in the field can benefit from it and also include the details of the future research scope.
5. There are few grammatical mistakes, read carefully and make necessary changes accordingly.
Author Response
- This paper had presented a simple and effective crack segmentation network with a novel scaling attention module.
Pros:
(1) The paper is easy to read and well organized, making it easier for readers to follow the authors idea and get a better understanding.
Author response
Thank you for your insightful review.
(2) Sufficient figures are provided to make the results more convincing.
Author response
Thank you for your insightful review.
Cons:
(1) Include a separate table mentioning the hyper-parameter settings of the proposed model.
Author response
We inserted table for hyper-parameter setting in section 4.3 Implementation Details.
(2) Mention the details pertaning to the model parameter uncertainty of the proposed model.
Author response
It is good point to mention and make clear that parameters in training affect prediction accuracy and model. we presented examples of trail parameters (batch size, learning rate, optimizer etc) to obtain the best prediction in section 4.3 Implementation Details.
(3) Discuss about the dice score stability during the training of the proposed model.
Author response
We mentioned dice score stability during training in section 4.3 Implementation Details.
(4) Make sure your conclusions appropriately reflect on the strengths and weaknesses of your work, how others in the field can benefit from it and also include the details of the future research scope.
Author response
We agree with your insightful comments. We should mention weakness of your work and future works. We mentioned them in section 5 conclusion.
(5) There are few grammatical mistakes, read carefully and make necessary changes accordingly.
Author response
Thank you for your comments, we carefully checked grammar mistakes again.
